# Ultraviolet spectroscopy of pressurized and supercritical carbon dioxide

Timothy W. Marin [1]✉ & Ireneusz Janik [2]✉

Carbon dioxide ($CO_2$) is prevalent in planetary atmospheres and sees use in a variety of industrial applications. Despite its ubiquitous nature, its photochemistry remains poorly understood. In this work we explore the density dependence of pressurized and supercritical $CO_2$ electronic absorption spectra by vacuum ultraviolet spectroscopy over the wavelength range 1455-2000 Å. We show that the lowest absorption band transition energy is unaffected by a density increase up to and beyond the thermodynamic critical point (137 bar, 308 K). However, the diffuse vibrational structure inherent to the spectrum gradually decreases in magnitude. This effect cannot be explained solely by collisional broadening and/or dimerization. We suggest that at high densities close proximity of neighboring $CO_2$ molecules with a variety of orientations perturbs the multiple monomer electronic state potential energy surfaces, facilitating coupling between binding and dissociative states. We estimate a critical radius of ~4.1 Å necessary to cause such perturbations.

---

[1] Department of Physical Sciences, Benedictine University, Lisle, IL, USA. [2] Notre Dame Radiation Laboratory, Notre Dame, IN, USA. ✉email: tmarin@ben.edu; ijanik@nd.edu

Understanding the physicochemical properties of carbon dioxide ($CO_2$) and its interaction with neighboring molecules is crucial for the control and modeling of a wide range of applications. Supercritical $CO_2$ is often used as a selective extractant in agribusiness, pharmacology, biochemistry, and material science. It is of practical import in that it replaces more hazardous, and often halogenated, organic extractants. In addition, due to its broadly tunable range of accessible densities, it presents many advantages as a potential coolant in Generation-IV nuclear reactor designs[1]. Consequently, thorough knowledge of supercritical $CO_2$ properties and $CO_2$ behavior under a variety of conditions benefits the design and control of processes taking place under supercritical conditions. Clearly, $CO_2$ is among the most important greenhouse gases responsible for terrestrial thermal regulation and thorough knowledge of its molecular absorption profile is necessary for accurate simulations of climate change. $CO_2$ was the dominant species in the atmosphere of Hadean Earth[2], contributing to its evolution towards the present Earth atmosphere. It now constitutes the majority of the Venusian and Martian atmospheres, and likely those of other exoplanetary bodies. $CO_2$ photochemistry must be understood over a wide range of excitation energies, as well as temperatures and pressures, in order to fully interpret the atmospheric chemistry of such worlds and to help assess its applicability as a potential nuclear power plant coolant.

The effect of temperature on the $CO_2$ absorption spectrum has been addressed in a number of previous studies[3–8], but the effect of pressure was not explored beyond sub-atmospheric levels. An increase of pressure/density of a subcritical or supercritical fluid increases the frequency of molecular collisions and simultaneously decreases the average distance between neighboring molecules. This alters the physicochemical properties of the fluid. For instance, the electronic perturbation of water monomers in supercritical water is evident upon increase of pressure/density, indicated by a gradual disappearance of diffuse vibrational structure and blueshift of its lowest-energy electronic absorption band[9]. However, distinction regarding whether the interaction between strongly H-bonding $H_2O$ monomers perturbs the ground state and/or the excited electronic state cannot be easily made. As interaction between neighboring $CO_2$ molecules upon condensation is arguably limited only to weak Van der Waals character, one may reason that with increased pressure/density a perturbation of the $CO_2$ upper electronic states should dominate and be easier to distinguish from effects occurring in the ground state.

The vacuum ultraviolet (VUV) absorption spectrum of $CO_2$ has been well studied experimentally[3,4,6–8,10–14]. Examining a compilation of the earliest works up to 1970[13], studies up to that point were incapable of resolving the rich spectral detail. That same publication demonstrated for the first time the diffuse vibrational structure inherent to the spectrum. Those studies were soon followed by reports that accurately obtained absorption cross-sections[12,14]. Later temperature-dependent studies were performed[3,4], examining a range of 200–370 K, which illustrated a significant increase in cross-section, especially in the long-wavelength region, with increasing temperature. In high-resolution experiments of the low-wavelength region, many discrete features were observed and tentative analyses of nine bands were provided, which defined an excited-state bending progression[10]. Detailed spectra obtained at lower wavelengths reported absolute cross-section measurements of $CO_2$ at 195 and 295 K in the wavelength region 1187–1755 Å[8,11]. These were later extended out to a wavelength of 2000 Å[5]. Within the last 7 years, two careful studies of the temperature dependence of $CO_2$ VUV absorption cross-sections have been performed[6,7]. These results extend previous studies[5,8] over a span of 150–800 K and curiously

show increasing absorption at long wavelengths with increasing temperature. All but the latest spectral studies have been summarized and are publicly available[15].

Representative $CO_2$ gas-phase data from the literature[5–8] in the range of 1200–2100 Å are shown in Fig. 1. The spectrum consists of two overlapped bands centered at 1480 and 1300 Å exhibiting diffuse vibrational structure. These weak absorption bands are due to electric dipole forbidden, but vibrationally allowed, transitions from the ground electronic state $^1\Sigma_g^+$ to a group of low-lying valence states (using customary electronic assignments for the $D_{\infty h}$ point group, appropriate for linear $CO_2$) $^1\Pi_g$, $^1\Sigma_u^-$, and $^1\Delta_u$. Two photodissociation channels associated with the 1480 Å and 1300 Å bands lead to the formation of $CO(X^1\Sigma^+)$ accompanied by either $O(^1D_0)$ or $O(^3P_j)$, respectively. It has been shown that a fully allowed channel leading to production of $O(^1D_0)$, with yields near unity close to maxima of absorption bands, is the major photolysis route over the 1200–1500 Å range[16,17]. Both $O(^1D_0)$ and $O(^3P_j)$ are produced at 1570 Å with the latter contributing ~6% to the overall yield[18]. It was soon deduced that this minor spin-forbidden photodissociation channel can be accessed by intersystem crossing from a bent singlet $^1B_2$ state to a triplet $^3B_2$ state (given a $C_{2v}$ point group representation)[19]. Other studies in the same wavelength range provided more details about rovibrational[20] or translational energy distributions of CO products[21], anisotropies in photodissociation channels, and their dependence on translational energies[22], as well as correlations of the CO vibrational-state distributions to the $O(^3P_{j=0,1,2})$ spin–orbit product[23]. The $O(^1D_0)$-producing channel aside, photodissociation studies below 1200 Å[24–26] provided additional insight into a $C + O_2$ photoproduct channel. In addition, studies near 1330 Å showed non-adiabatic photodissociation dynamics of a channel leading to $O(^3P_j)$ production[27]. The nature of the diffuse vibrational structure, absorption cross-sections, and transition assignments of the two broad continuum bands in relation to $CO_2$ photodissociation channels were the subjects of a number of theoretical studies over the years[28–36]. It has been concluded that manifold potential energy surfaces (PESs) can be ascribed to the observed VUV photoexcitation.

The low-energy band over the range 1383–2000 Å shows diffuse but weak vibrational structure. Maximum absorption cross-sections of $6.68 \times 10^{-19}$ cm$^2$ (175 M$^{-1}$ cm$^{-1}$) at 1459 Å[11] or $6.46 \times 10^{-19}$ cm$^2$ (169 M$^{-1}$ cm$^{-1}$) at 1457 Å[8] were both reported. The state was assigned as $^1\Delta_u$, corresponding to a $2\pi_u^* \leftarrow 1\pi_g$

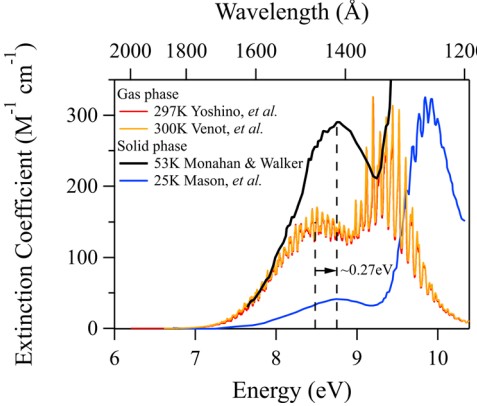

**Fig. 1 High-resolution gas-phase and solid-phase $CO_2$ VUV transmission spectra.** The gas-phase (red, orange)[5,8] spectra clearly illustrate broad continuum bands with overlapping diffuse structure absent from the solid-phase spectra (blue, black)[42,43,47]. Vertical dashed lines indicate the spectral peaks to illustrate the 0.27 eV shift of the solid-phase vs. gas-phase $CO_2$ spectrum.

electronic transition[13]. The vibrational structure in this wavelength region was believed to correspond to a bending mode with a frequency of ~544 cm$^{-1}$. In a multi-reference configuration interaction calculation, the PESs of the interacting $^1\Sigma_u^-$ and $^1\Delta_u$ states were calculated, with the former lying lower in energy[34]. The interaction of these two states was deemed highly important for the understanding of the diffuse vibrational structures in this energy region[21,37,38]. Indeed, six different excited electronic states can be accessed at energies below ~8.5 eV in the Franck–Condon region of the absorption spectrum—three singlet and three triplet ($2^1A'$, $1^1A''$, $2^1A''$ and $1^3A'$, $1^3A''$, $2^3A''$, respectively, in a $C_s$ point group representation)[36]. The $^1A''$ and $2^1A''$ states couple by non-adiabatic coupling matrix elements (NACME)[29,30,36]. Transition dipole moments to the $2^1A''$ state are small and its contribution to spectrum is typically omitted in theoretical analysis[36]. For an undistorted, linear molecule, the three singlet states have similar energies and mutually couple by Renner–Teller pairing and NACME[28,33,36]. For photon energies less than ~8 eV, absorption primarily takes place upon decrease of the O–C–O bending angle below 170°[36]. Alternatively, it has been proposed that transitions below an energy of 8.5 eV have non-vertical (non-Franck–Condon) character from considerable shift along the bending angle during excitation[31]. The PESs of the ground-state $1^1A'$ (X) and singlet excited $2^1A'$ state can cross at a conical intersection (CI) for a bond angle of ~100° when both C–O bond lengths are equivalent (as both states belong to different irreducible representations of $C_{2v}$ configuration). In a $C_s$ configuration, the potentials form an avoided crossing between X and $2^1A'$. Therefore, near the CI both states are coupled by NACME and their respective vibrational states are mixed. The low-energy part of the measured absorption spectrum reflects this mixing in the form of irregular sub-nanometer-wide spectral features. The authors make use of an autocorrelation function as the scalar product of the initial wavepacket at time $t = 0$ and the evolving wavepacket at time $t$, establishing the interrelation between the time and the frequency domain. It is the link between the time-dependent molecular dynamics and the absorption spectrum; the spectrum is the Fourier transform of the autocorrelation function, where (0,0,0) represents the system coordinates prior to excitation. The wider undulations apparent for $CO_2$ in the lower-energy band with a spacing of about 520 cm$^{-1}$ at 1500 Å and 650 cm$^{-1}$ at 2000 Å correspond to a recurrence of the (0,0,0) autocorrelation function with a period of ~60 fs and were largely reproduced by calculations applying wavepacket methodology[35,36]. Alternatively, the nature of such undulations was associated with bending excitations in the states $2^1A'$ and $1^1A''$, and its irregularity has been interpreted as a manifestation of the carbene-type "cyclic" O–C–O minimum[28]. The second high-energy $CO_2$ absorption band shows sharp, more regular structure between 1200 and 1383 Å. Maximum cross-sections of $1.22 \times 10^{-18}$ cm$^2$ (319 M$^{-1}$ cm$^{-1}$)[11] and $1.25 \times 10^{-18}$ cm$^2$ (327 M$^{-1}$ cm$^{-1}$)[8] at 1349 Å were both reported. The excited state was assigned as $^1\Pi_g$, corresponding to a $3s\sigma_g \leftarrow 1\pi_g$ electronic transition. This is the lowest singlet Rydberg state, which retains valence character[39]. The vibrational structure with an average spacing of ~600 cm$^{-1}$ corresponds to a bending mode. The absorption in this wavelength region is a $^1B_2$ component of a mixed Rydberg-valence transition with irregular vibrational structure. The interaction of vibrational and rotational angular momenta was thought to be the cause of the irregularities[13]. It has been found that they are due to the crossing of two nearly degenerate linear states $^1\Delta_u$ and $^1\Sigma_u^-$ with the $^1\Pi_g$ state, resulting in a CI in the Franck–Condon region of the absorption spectrum[34].

Very detailed analysis of $CO_2$ absorption cross-sections from first principles has been the subject of recent computational works[31]. A model combined high level ab initio PESs of four excited electronic states and their coordinate-dependent transition dipole moment vectors with quantum mechanical calculations of the absorption spectra, corresponding to excitations from many initial rovibrational states of $CO_2$. Calculations reliably predict absorption cross-sections for spectra up to 2500 Å and temperatures up to 2500 K, agreeing well with recent experimental spectra[6,7]. With increasing temperature, ground-state vibrational population spreads along the bending angle towards the bent wells of $2^1A'$ and $1^1A''$ states due to increased available thermal energy. Hot band absorption appears along the red edge of the VUV spectrum due to the increased population in these excited vibrational levels. The study identifies that the symmetric stretch and the bending mode (strongly coupled via a 1 : 2 Fermi resonance) in the ground state are responsible for this observed increased long-wavelength absorption at elevated temperatures.

There are few measurements of $CO_2$ VUV absorption spectra in condensed phases. Early near-ultraviolet spectroscopic studies of liquid $CO_2$ from $-51$ °C to room temperature found an absence of absorption from 2150 Å to 6000 Å[40], contrasting preceding erroneous results that implied the existence of absorbing complexes within that spectral range[41]. VUV absorption data in the solid phase were first reported over four decades ago[42,43] and, more recently, by a number of groups studying interstellar ice analogs[44–47]. It was shown that spectral interpretation of absorption in thin films of ices can be affected by errors arising from the accumulation of photodissociation products in the bulk, obscuring band assignment. Further complications arise from photodesorption, which affects the measured values of absorption cross-sections. Upon solidification, the $CO_2$ VUV absorption spectrum[42,43,47] loses the low-energy diffuse vibrational structure observed in the gas phase, but maintains its characteristic two broad bands for the $^1\Delta_u \leftarrow {}^1\Sigma_g^+$ and $^1\Pi_g \leftarrow {}^1\Sigma_g^+$ transitions, although they are blue shifted by 0.3 and 0.6 eV to 8.8 and 9.9 eV, respectively (see Fig. 1). This large shift is attributed to a perturbation of the Rydberg orbital[48], which in the solid phase exhibits extensive vibrational structure with a mean separation of ~609 cm$^{-1}$ between features.

$CO_2$ is entirely compressible as a supercritical fluid (critical temperature and pressure = 304.1 K, 73.8 bar), so by tuning the sample pressure, a variety of densities can be accessed. Changing density alters the proximity of nearest-neighbor molecules. For spectroscopic studies as a function of density, this implies the ability to gain direct insight on the influence of intermolecular forces on spectroscopic features. How density affects the structure of supercritical $CO_2$ as it undergoes changes from a low-density gas-like state to a higher-density liquid-like state was examined by Raman spectroscopy[49]. Changes in pressure induced a change in the Fermi resonance between the fundamental symmetric stretch $\nu_1$ and first overtone of the bending vibration $2\nu_2$. With a gradual increase of density from 0.029 g cm$^{-3}$ (gas-like) to 0.44 g cm$^{-3}$ (liquid-like), the wavenumbers of both vibrational frequencies were found to linearly decrease by several cm$^{-1}$, indicating relatively weak interaction between $CO_2$ molecules in the ground state.

With these Raman studies[49] in mind, our current pressure/density-dependent VUV absorption studies should give direct insight into the interaction of $CO_2$ molecules in the excited state. We present VUV absorption spectra for supercritical $CO_2$ acquired over the wavelength region 1445–2000 Å (8.58–6.20 eV) at 308 K with a resolution of 5 Å. Spectra were measured at multiple pressures over the range 19–137 bar, corresponding to densities of 0.036–0.767 g cm$^{-3}$. With increasing density, we find a gradual quenching of the diffuse vibrational structure inherent to the gas-phase spectrum, which we analyze in terms of perturbation of the monomer electronic structure. We compare these observations to a similar effect recently reported for supercritical $H_2O$[9].

## Results and discussion

**$CO_2$ VUV absorption spectra and density dependence**. The high-pressure and supercritical $CO_2$ VUV spectra acquired at 308 K are shown in Fig. 2. Densities associated with each spectrum are listed in Table 1. Despite the low 5 Å resolution of these experiments, the known diffuse vibrational structure is still apparent in the low-pressure limit and the observed wavelength of maximum absorption at 1460 Å agrees with previous measurements performed at temperatures in the range of 295–300 K. It is clear that our data compare particularly well with the previous gas-phase results acquired at room temperature[5–8] (Fig. 3), despite our inferior wavelength resolution and signal-to-noise ratio. The most recent temperature-dependent studies[6,7] suggest that with increasing temperature we should expect a small redshift of our $CO_2$ spectrum. Comparing those data acquired at 300 K to the older results obtained at 295 K[5,8] (Figs. 1 and 3) illustrates that any redshift is difficult to distinguish for such small differences in temperature. Therefore, it is understandable that our data at 308 K compare quite well with previous work at these slightly lower temperatures[5–8].

Examining Fig. 2, it is apparent that with increasing pressure the diffuse vibrational structure is gradually quenched and by a pressure of 90 bar ($0.595 \, \text{g cm}^{-3}$) the remaining structure is nearly lost within the signal-to-noise ratio. Surprisingly, no apparent spectral energy shift is observed with increasing pressure. In an analogous VUV experiment performed in supercritical $H_2O$, we observed a 0.32 eV blueshift upon increasing density up to $0.527 \, \text{g cm}^{-3}$ (296 bar), which we attributed to an increasing extent of hydrogen bonding and creation of a dynamic equilibrium between $H_2O$ monomers and oligomers (dimers, trimers, etc.)[9]. The spectral shift was related to lowering of the ground-state energy upon dimerization/oligomerization (condensation) and hydrogen bond formation. In the extreme case, for hexagonal $H_2O$ ice, the absorption band is observed to blueshift by 1.16 eV compared to the gas phase.

One might assume a similar shift of the $CO_2$ ground-state energy and VUV absorption spectrum upon dimerization and condensation, although likely to a lesser extent due to relatively weak van der Waals interactions compared to the hydrogen bonding present in $H_2O$. Indeed, the observed spectrum for $CO_2$ ice (density = $1.60 \, \text{g cm}^{-3}$) blueshifts by 0.27 eV compared to the gas phase (see Fig. 1)[47]. Hence, one might conclude that the interaction energy should be at least $1.16 \, \text{eV} / 0.27 \, \text{eV} = 4.3$ times smaller than the hydrogen-bonding forces in $H_2O$. Naively assuming that the extent of interactions in $CO_2$ scales linearly with pressure/density, we could speculate that for increasing the pressure and density up to 90 bar and $0.595 \, \text{g cm}^{-3}$ (37% of the density of $CO_2$ ice or of liquid $CO_2$ near its triple point) the spectrum might shift by 37% of the 0.27 eV difference between the gas- and solid-phase spectra, or a blue shift of merely 0.10 eV (~17 Å). No such shift is apparent in our data when comparing the spectra obtained at 90 bar or even up to 137 bar (Fig. 2), although it should be easily detectable given our wavelength resolution. Clearly, intermolecular forces playing a part in causing the energy difference between the gas-phase and solid-phase $CO_2$ spectra, manifesting in a spectral shift, must require even higher densities and therefore smaller distances between molecules at which collective forces might come to play as well. To conclude, the interactions between $CO_2$ molecules are demonstrably far weaker than in $H_2O$ and must require much shorter interaction distances to become observable.

We note that we are currently constructing a new in-house VUV spectrometer that eliminates the need to perform these experiments in conjunction with a synchrotron light source and will also allow us to access higher pressures. This instrument will allow exploration of the liquid $CO_2$ spectrum near its triple point and densities comparable to the solid phase. This instrument will

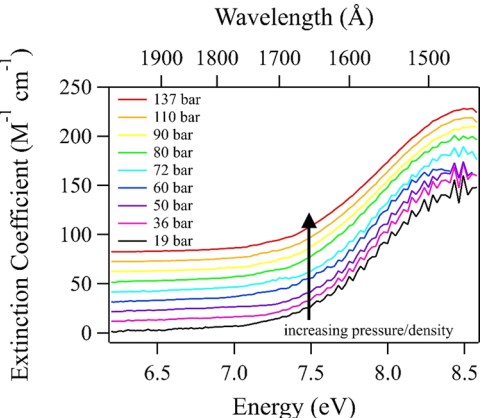

**Fig. 2 Supercritical $CO_2$ VUV spectra.** Spectra were acquired at 308 K at pressures/densities as listed in Table 1. Spectra are artificially offset in increments of $10 \, \text{M}^{-1} \text{cm}^{-1}$ for viewing.

**Table 1 Pressures and corresponding densities for all supercritical $CO_2$ spectra acquired, and estimated average intermolecular radii $\langle r \rangle$, assuming ideal gas behavior.**

| Pressure (bar) | Density ($\text{g cm}^{-3}$) | Intermolecular radius (Å) |
|---|---|---|
| 19 | 0.036 | 6.80 |
| 36 | 0.077 | 5.44 |
| 50 | 0.121 | 4.68 |
| 60 | 0.164 | 4.23 |
| 72 | 0.246 | 3.70 |
| 80 | 0.418 | 3.10 |
| 90 | 0.595 | 2.75 |
| 110 | 0.695 | 2.62 |
| 137 | 0.767 | 2.53 |

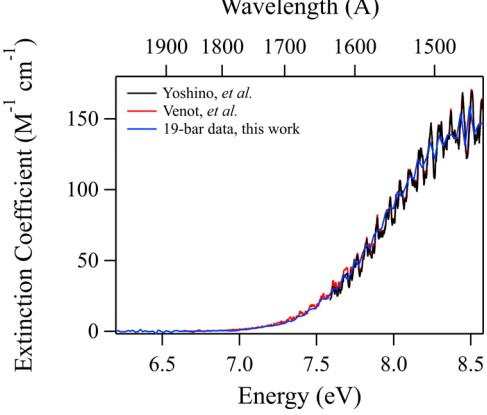

**Fig. 3 Comparison of long-wavelength 19 bar supercritical $CO_2$ spectrum (308 K) to other results in the literature.** Results are compared to those of Yoshino et al.[5,8] (295 K, blue) and Venot et al.[6,7] (300 K, red).

allow us to examine formation of the liquid phase to ascertain whether there is a sudden change in the spectrum position, band shape, or band features. We likewise propose to investigate the pressure/density dependence of gas-phase $CO_2$ at lower temperatures to confirm the existence of similar gradual loss in vibrational structure with increasing pressure.

As determining the absolute peak wavelength of a band with rich diffuse vibrational structure has some element of subjectivity,

we fit our spectra with a simple Gaussian profile to look for any trends in spectral shape or position upon pressurization. Fitting a simple Gaussian profile does reveal a small increase in bandwidth with increasing pressure; a standard deviation of 0.70 eV is observed at 19 bar, which increases to 0.79 eV at 137 bar. As we could record only the lower-energy part of the diffuse vibrational band due to the onset of the sapphire window absorption in the sample cell, it might be argued that the observed broadening upon pressurization actually corresponds to a redshifting of the band. The assumed extent of redshift based on the difference between halfwidths of fitted Gaussian profiles comes to about 0.11 eV. One can apply similar Gaussian fits to the reported spectra in the 150–800 K range[6,7] and relate the observed redshift and broadening with increasing temperature to our own data. Doing so suggests that a temperature increase of about 142 K is needed to generate a 0.11 eV redshift in our data, which is entirely implausible. Thus, it is highly unlikely that the absorption band is redshifting with increased pressure/density.

Temperature-induced spectral broadening is associated with increased ground-state vibrational excitation[32], which causes an increase in the Franck–Condon factors for transitions to the $2^1A'$ and $1^1A''$ states. It may be that the Franck–Condon factors for these transitions are pressure-dependent even though the Boltzmann factors should not be affected in our constant-temperature experiments. One could speculate that with increased collision frequency at higher densities, a larger population of ground-state $CO_2$ can achieve a bent geometry, and hence for that population we observe an increase in the Franck–Condon factors, reflected by increased intensity on the low-energy side of the absorption band. In addition, what should be noted from recently reported temperature-dependent data[6,7] is that an increase of temperature up to even 800 K does not extinguish the diffuse vibrational structure of the first absorption band. Seemingly, this is only a consequence of the high-pressure/density conditions in our data.

We considered the possibility of collisional/pressure broadening in reducing the diffuse vibrational structure with increasing pressure. Clearly, such effects should become more important with increasing pressure. At the highest pressure/density for which spectra were recorded (137 bar, 0.767 g cm$^{-3}$), a textbook hard sphere collisional broadening model predicts a broadening of <20 cm$^{-1}$ (< 2 meV) in magnitude. Applying a simple Lorentzian convolution that is 20 cm$^{-1}$ full width at half maximum to our 19 bar (i.e., lowest pressure/density) data, where collisional/pressure effects are least significant, negligibly affects the observed vibrational structure. Therefore, collisional broadening cannot justify the disappearance of spectral detail at higher pressures in our data.

**Consideration of dimer formation**. Our recent studies of supercritical water demonstrated that the effects of dimer formation can occur at distances far exceeding the equilibrated dimer geometry[9]. At the small distances between molecules in the supercritical domain compared to the gas phase, an array of possible transient metastable $H_2O$ dimer geometries was considered, which can alter both the ground- and excited-state energies, thus creating "electronically perturbed" monomers. The perturbation of $H_2O$ monomers in the ground state can affect the relative position of the Franck–Condon region reflected on the excited-state PES. If the monomer PES is distorted by interaction with a nearby molecule, the wavepacket trajectory ascribed to the symmetric stretch no longer stays on the rim of the dissociative PES between the two HO–H dissociation channels. In addition, perturbation of the $H_2O$ excited state can further increase the already unstable trajectory on top of the saddle of the

PES, for instance by narrowing the rim between two dissociation channels and damping the oscillatory motion. Either case would prohibit wavepacket oscillation in the inner region from recurring to its position of origin and rather lead to a dissociation channel. Effects such as these would be capable of quenching the diffuse vibrational structure inherent to the true $H_2O$ monomer spectrum.

Unlike $CO_2$, the $H_2O$ gas-phase VUV absorption spectrum for the lowest-lying electronic state and observed undulations is related to the nature of a single dissociative electronic state, with electronic symmetry $^1B_1$ in a $C_{2v}$ configuration. The origin of the diffuse vibrational structure for the lowest-lying absorption band of $CO_2$ is more complex and related to six overlapping excited states and their collective contributions to the indirect photodissociation dynamics. Following assignments that have been presented in the literature[36], in our recorded spectral range (1455–2000 Å) near the peak, the transition $1^1A' \rightarrow 2^1A'$ contributes ~90% to the overall spectral intensity, complemented somewhat by the $1^1A' \rightarrow 1^1A''$ transition. The latter gradually contributes more with increasing wavelength and shows equivalent contribution to the $1^1A' \rightarrow 2^1A'$ transition at 2000 Å. However, at this wavelength our recorded extinction is near zero. Therefore, for the sake of simplicity we limit our discussion to pressure effects arising just from the $1^1A' \rightarrow 2^1A'$ transition. Considering that we cannot observe sub-nanometer spectral structure due to our limited spectral resolution, we cannot discuss how pressure affects the low-energy part of the spectrum where the ground-state $1^1A'$ and excited-state $2^1A'$ mix in their respective manifolds of vibrational states (near 100° bending angle, in the vicinity of the CI)[36]. We are more concerned with the disappearance of the spectral undulations with a spacing of about 650 cm$^{-1}$ at 2000 Å and 520 cm$^{-1}$ at 1500 Å, related to recurrence of the aforementioned (0,0,0) autocorrelation function with a period of ~60 fs. These undulations arise mainly from bending motion in the deep well of the $2^1A'$ PES (modulated by motion in the deep well of the $1^1A''$ PES, apparent towards longer wavelengths), at bent geometries above and below the singlet channel threshold[35]. Assuming a relatively low effect of density on the ground state (see below), judging by the small pressure effect on Raman spectra of supercritical $CO_2$[49], we can expect that perturbation of $CO_2$ monomers would occur mostly by affecting the PESs of intersecting singlet and triplet excited states. The apparent disappearance of such undulations would indicate that upon perturbation, large amplitude vibrational motions in the singlet electronic state $2^1A''$ ($1^1A''$) normally leading to recurrences in the autocorrelation function would become extinguished because of an increase in already existing coupling to the dissociative triplet states $1^3A''$ and $2^3A''$. Computational studies show that dissociation products (CO and O) can be trapped in a ~1.9 eV deep potential well apparent in a two-dimensional representation of a $2^1A'$ ($1^1A''$) PES. Pressure perturbation of all PESs in excited states can either lead to the change of the depth of singlet state potential well and/or affect the position of the seams of intersections of that PES with the triplet states PESs. As there are six excited states that could potentially be affected by such perturbations vs. a single ground state, we conjecture that excited-state electronic perturbation is most likely responsible for the disappearance of diffuse vibrational structure in the VUV spectrum upon increase of pressure. Yet, the effect on the ground state is worth evaluation as well.

To estimate the extent of interactions in the ground state, we considered the possibility of dimer formation and the corresponding possible spectral contribution with increasing pressure/density. Previous theoretical studies considered multiple possible dimer geometries (T-shaped, parallel, crossed, and slipped parallel), all resulting in considerably different bound energies

ranging from <20 cm$^{-1}$ to >500 cm$^{-1}$, and equilibrated inter-molecular distances $R$ ranging from 3.26 Å to 6.20 Å[50,51]. The global minimum is found for a slipped parallel 60° arrangement with a bound energy of 515 cm$^{-1}$ and $R = 3.53$ Å. Hence, a variety of different dimer geometries could conceivably provide a distribution of dimer energies and shift the electronic transition energy by varying amounts, effectively smearing out the vibrational structure. The known $CO_2$ equilibrium constant predicts <8% dimers even at the highest pressure measured[52]. Therefore, even if dimer formation were to cause a spectral energy shift, the observed spectrum would be a weighted average of dimer and monomer contributions, still almost entirely weighted towards the monomer spectrum. It is unlikely that we would detect this in our experiment given our low-wavelength resolution, confirming what we already discussed.

The increased coupling of singlet to triplet states upon pressurization can be quantified by the extent of diffuse vibrational structure present in each of our supercritical $CO_2$ spectra. As already mentioned, the spectrum backbone can be fit well by a single Gaussian function. Subtracting this fit leaves a residual containing the vibrational detail, as seen in Fig. 4. It is noteworthy that we carried out the fit only down to an energy of 7.1 eV, below which the vibrational detail is no longer apparent. After taking the absolute value of the residual, one can sum the data points for any spectrum at a given pressure/density to obtain its total magnitude. The result is shown in Fig. 5. The extent of loss of the spectral detail with increasing density follows an exponential decrease. We extrapolate back to a density of 0 to impose a fraction of vibrational structure that would be equal to 1 in the limit of $CO_2$ monomer.

**Estimation of critical interaction radius.** The presence of a second $CO_2$ molecule nearby will almost certainly perturb symmetry and electronic structure, but how close does the second molecule need to be? We can estimate a critical radius, $r_c$, for quenching the spectral diffuse vibrational structure, guided by the exponential behavior illustrated in Fig. 5. Assuming ideal gas behavior and completely random distances, the probability $P_i$ to find $i$ $CO_2$ molecules within distance $r_c$ of another will be given by Poisson statistics as

$$P_i = \frac{N_s^i}{i!} e^{-N_s} \tag{1}$$

where given the average number density, $\rho$, the average number of $CO_2$ molecules within the spherical critical volume, $N_s$, is given by

$$N_s = \frac{4\pi r_c^3 \rho}{3} \tag{2}$$

To observe the greatest extent of spectral diffuse vibrational structure, we need to have $i = 0$ $CO_2$ molecules within $r_c$, so Eq. (1) reduces to $P_0 = e^{-N_s}$ and a simple exponential behavior with increasing density is recovered. Based on Fig. 5, a mass density of 0.224 g cm$^{-3}$ ($\rho = 0.00307$ Å$^{-3}$, achieved at $p = 70.4$ bar) corresponds to the point where the magnitude of vibrational structure has been reduced to $1/e$ of its maximum value. By the analysis above, $r_c$ must be 4.27 Å—a benchmark for the expected value in the absence of intermolecular interactions. The mean intermolecular distance $\langle r \rangle$ is given by

$$\langle r \rangle = \int_0^\infty 4\pi r^2 \rho r \exp\left(-\frac{\rho 4\pi r^3}{3}\right) dr \tag{3}$$

This works out to be $\langle r \rangle = 3.81$ Å at this density. $\langle r \rangle$ is listed for all our experimental pressures/densities in Table 1.

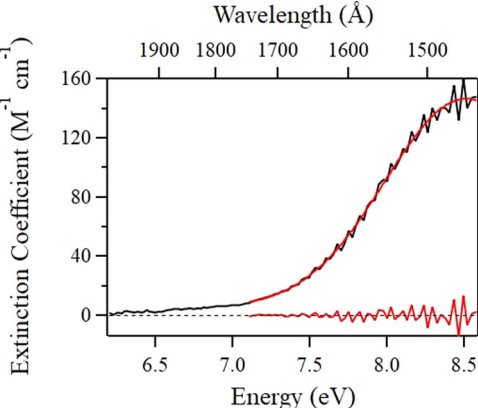

**Fig. 4 Supercritical $CO_2$ spectrum at 19 bar and fit.** Experimental data at 0.036 g cm$^{-3}$ are shown (black curve) along with Gaussian fit (red curve). The resulting residual can be seen at the bottom of the graph. The dashed black line is placed merely to guide the eye through a vertical intercept of 0.

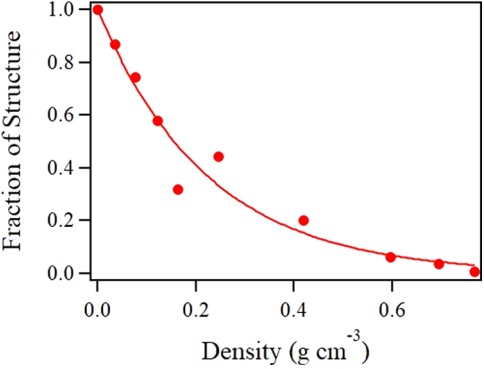

**Fig. 5 Fraction of diffuse vibrational structure in the $CO_2$ monomer spectrum.** The extent of vibrational structure inherent to the $CO_2$ spectrum shows an exponential decrease as a function of increasing density for supercritical $CO_2$.

We note that a similar analysis was conducted for supercritical $H_2O$ in our prior work[9], giving $r_c = 5.49$ Å, using an ideal gas model, and $r_c < 4.5$ Å based on a high-quality $H_2O$ dimer potential. It shows that using the ideal gas approximation in the Poisson statistics for a strongly interacting species such as $H_2O$ can overestimate the critical distance by 20% or more. Therefore, we turned to the high-quality PES for the $CO_2$ dimer[50,51] to estimate $r_c$ for several dimer geometries, calculated from a CCSD(T)-F12 level of theory in connection with an augmented correlation-consistent aug-cc-pVTZ basis set. The dimer potential was calculated out to at least $r = 20$ Å for the slipped parallel (both 60° and 45° orientations), linear, crossed, and T-shaped geometries described by the authors, using codes that they kindly supplied[51]. These potentials are illustrated as a function of $r$ in Fig. 6. A fifth-order polynomial expansion of a Lennard–Jones potential, $U(r)$, was used to fit each function, achieving excellent agreement. The running coordination number $N(r)$,

$$N(r) = \int_0^r 4\pi r^2 \rho \exp\left(-\frac{U(r)}{RT}\right) dr \tag{4}$$

was then calculated as a function of distance at each experimental density, as well as at a density of 0.224 g cm$^{-3}$, where $1/e$ of the vibrational structure remains. For comparison, a hard sphere

potential was also used to calculate $N(r)$, using a kinetic diameter of 3.30 Å for $CO_2$. The intermolecular critical distance $r_c$ required to give the fraction of vibrational structure quenched at each density, per Fig. 5, is obtained when $N(r)$ is equal to that fraction. Representative results are shown in Fig. 7 for the slipped parallel 60° geometry at three densities, alongside similar results for a hard sphere model. In short, the attractive potential associated with this particular dimer potential reduces $r_c$ to 3.38 Å from the ideal gas value of 4.27 Å when $1/e$ of the vibrational structure remains. For comparison, the 4.40 Å hard sphere value is slightly larger than the ideal gas value, owing to the kinetic diameter accounting for the molecular volume neglected by the ideal gas estimation.

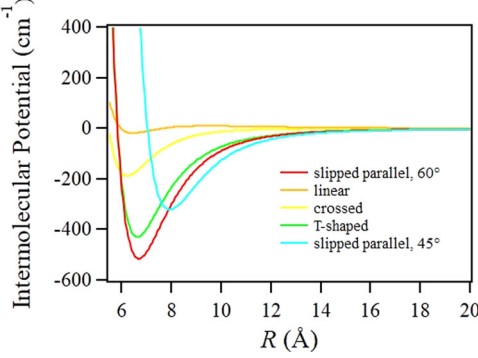

**Fig. 6 $CO_2$ dimer intermolecular potentials.** Dimer potentials are illustrated as a function of $CO_2$ carbon–carbon distance for several dimer geometries based on a high-quality four-dimensional $CO_2$ dimer potential energy surface[51].

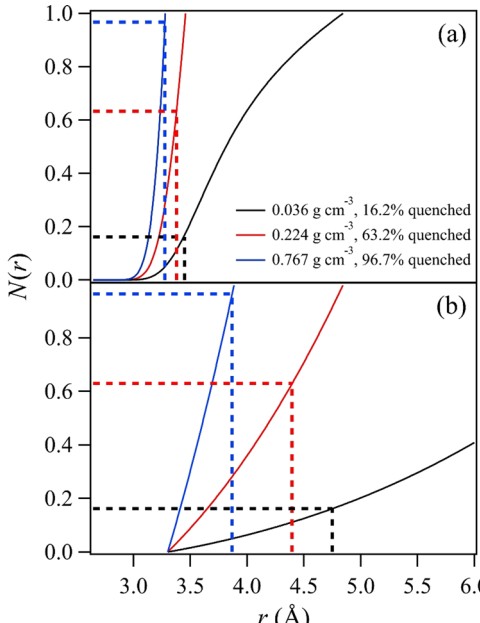

**Fig. 7 Running coordination number for $CO_2$ dimers.** The running coordination number is shown as a function of carbon–carbon distance at the lowest experimental density (0.036 g cm⁻³) and highest density (0.767 g cm⁻³) measured, as well as at 0.224 g cm⁻³, the density associated with $1/e$ of the vibrational structure inherent to the spectrum remaining. Horizontal lines indicate the fraction of vibrational structure quenched at a particular density and vertical lines indicate the associated critical radius. **a** Data for the slipped parallel 60° geometry and **b** data for a hard sphere treatment.

Figure 8 shows $r_c$ across all experimental densities, as calculated for several dimer geometries. With the exception of the linear geometry, which has a slightly repulsive potential at distances longer than ~4.00 Å, all dimer geometries produce a value of $r_c$ that is substantially less than that produced by an ideal gas or hard sphere model. Table 2 summarizes the results at 0.224 g cm⁻³; the average value across all dimer potentials is 4.14 Å and we estimate an uncertainty of ±0.2 Å in our analysis. We note that our estimated value of $r_c$ is significantly larger than the equilibrium dimer separation distance of 3.53 Å. Hence, our calculated critical radii predict significant attractive intermolecular forces, but not enough to bring about a prevalence of equilibrated dimer formation. This is understandable, considering the attractive component of the potential approximately scaling as $r^{-6}$; the force of attraction does not become substantial in magnitude without very close proximity of $CO_2$ molecules. It is interesting to note the spatial dimensions for solid $CO_2$, for which one could argue that attractive intermolecular forces are maximized. The Pa3 cubic unit cell dimension for $CO_2$ containing four molecules is 5.642 Å[53]. By Eq. (3), this gives an average nearest-neighbor distance of $\langle r \rangle = 1.97$ Å.

Comparing to supercritical $H_2O$, we note that the $H_2O$ $1/e$ density was 0.0432 g cm⁻³, over a factor of 5 less in mass density than what we observe for the $CO_2$ $1/e$ density value of 0.224 g cm⁻³. However, comparing number density, the difference is just over a factor of 2 (0.00144 vs. 0.00307 Å⁻³ for $H_2O$ and $CO_2$, respectively). The implication is that, compared to $H_2O$, $CO_2$ requires over five times the amount of matter (or over twice the number of molecules) packed into the same volume to achieve

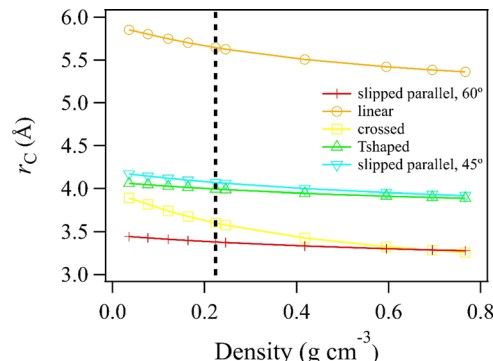

**Fig. 8 Predicted critical radii for $CO_2$ dimer geometries.** Critical interaction radii are shown at each experimental density for slipped parallel (both 60° and 45° orientations), linear, crossed, and T-shaped geometries described per the $CO_2$ dimer potential energy surface[51]. The black vertical line indicates a density of 0.224 g cm⁻³, where $1/e$ of the vibrational structure inherent to the spectrum remains.

**Table 2 Predicted critical radii $r_c$ associated with a density of 0.224 g cm⁻³, where $1/e$ of the vibrational structure inherent to the $CO_2$ monomer spectrum remains, for various models applied.**

| Model | $r_c$ (Å) |
| --- | --- |
| Ideal gas | 4.27 |
| Hard sphere | 4.40 |
| Slipped parallel, 60° | 3.38 |
| Linear | 5.64 |
| Crossed | 3.60 |
| T-shaped | 3.99 |
| Slipped parallel, 45° | 4.07 |

a similar extent of electronic perturbation. This demonstrates consistency with the contrasting intermolecular forces between the two species. Considering the lack of dipole or hydrogen-bonding capabilities for $CO_2$, a much closer proximity of molecules is perhaps entirely expected to achieve enough electrostatic influence between molecules that electronic structure can be altered.

## Conclusions

VUV electronic absorption spectra of pressurized and supercritical $CO_2$ were measured over the wavelength range 1455–2000 Å. We focused on examining spectral changes with increasing pressure/density (range 19–137 bar, 0.036–0.767 $g\,cm^{-3}$) at 308 K, just above the critical temperature of 304 K, for the lowest-lying electronic absorption band. Although we observed no pressure effect on the band peak energy position, the known diffuse vibrational structure inherent to the gas-phase VUV absorption spectrum gradually decreases in magnitude with increasing pressure/density. At the highest densities measured, it is nearly unobservable, yet the small amount of observed broadening with increasing density may indicate a corresponding increased population of ground-state $CO_2$ molecules having bent geometries.

After discarding collisional broadening and dimerization as potential causes of the loss of spectral detail, we discussed the observed changes in the context of what is known as quite complex $CO_2$ electronic structure[28–36]. With increasing pressure/density, the close proximity of neighboring $CO_2$ molecules may perturb the multiple monomer upper electronic PESs, leading to enhanced coupling between binding and dissociative states, and eliminating the wavepacket recurrence that gives rise to the diffuse vibrational structure. Based on a high-quality $CO_2$ dimer PES[51], we estimated a critical radius of 4.1 ± 0.2 Å between molecules necessary to cause such perturbations, which could arise from a distribution of non-equilibrated intermolecular orientations and geometries.

Our current development of an in-house VUV spectrometer with high-pressure capabilities will allow us to extend this work to pressure and temperature conditions near the $CO_2$ triple point, to observe spectral changes when condensation becomes thermodynamically feasible. In accessing high pressures, we can achieve densities approaching that of liquid and solid $CO_2$ to further probe any potential spectral shifts or change in band shape when interaction with neighboring molecules is expectedly stronger than for the currently presented work. Similarly, exploration of pressure dependence of gas-phase spectra under such conditions may show stronger evidence of dimerization.

## Methods

Absorption spectra were measured using a synchrotron-based experiment. The unique nature of the high-sensitivity, high-pressure, high-temperature capable VUV experiment and technical details regarding the light source and detection, sample cell specifications, sample preparation, temperature/pressure control, and sample flow have been published previously[9,54,55]. Supercritical fluid extraction grade carbon dioxide was obtained from Praxair (POL545). The carbon dioxide cylinder was attached directly to the sample flow system without a pump, as the inherent pressure in the cylinder effectively causes sample to flow through. However, in this manner, the maximum pressure is limited to the pressure in the cylinder. Lower pressures were set using a back pressure regulator (54-2162T24, Tescom). VUV measurements were carried out at the Stainless Steel Seya beam line of the Synchrotron Radiation Center, University of Wisconsin–Madison. The combination of an adjustable path length sample cell, synchrotron light source, and secondary filtering monochromator on the beam line allowed for six orders of magnitude in light detection dynamic range. Photons were generally counted for 1 s per data point and photon counts were normalized to fluctuations in the synchrotron beam current in real time during data acquisition. When transmittance was low (<$10^4$ counts $s^{-1}$), photons were counted for up to 20 s per point. Typically, 30 min were necessary to acquire an entire spectrum with a wavelength resolution of 5 Å. The beam line monochromator, secondary filtering monochromator, and photon counter were controlled and synchronized through Igor

Pro 6.0 run on a notebook PC. The reported spectra are actually composite spectra, compiled from two sample cells with different path lengths to accommodate measuring six orders of magnitude in absorbance/extinction. Temperature and pressure conditions during measurements were stable within ±0.2 K and ±0.2 bar, respectively. Measurements were conducted at 308 K, at the pressures/densities listed in Table 1. Spectra were acquired using gold foil spacers in the sample cell to designate a path length of 10 μm (for measurements near the absorption maximum) or 91.4 μm (for measurements on the absorption tail). The blue edge cutoff of the spectra is due to the onset of the sapphire window absorption.

## Data availability

The datasets generated during and analyzed during the current study are available from the corresponding authors on reasonable request.

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

## Acknowledgements
This work is based in part upon research conducted at the Synchrotron Radiation Center, University of Wisconsin–Madison, which was supported by the National Science Foundation (NSF) under Award DMR-0537588. We thank the SRC staff, in particular G. Rogers, M. Bissen, M. Severson, M. Fisher, G. Vlasak, and D. Wallace. The help from machine shop staff of the Physics Department at University of Notre Dame is greatly appreciated. We thank K. Darr and M. Richmond from the Notre Dame Nanofabrication Facility for introduction to and help in metal vapor deposition techniques and instrumentation. We thank J. Kalugina for the sharing her code needed for calculating the $CO_2$ dimer potential energy surface. We also thank J.B. Nee for sharing the experimental data for the $CO_2$ absorption spectra measured by his group. The research described herein was supported by the Division of Chemical Sciences, Geosciences, and Biosciences, Basic Energy Sciences, Office of Science, United States Department of Energy, through grant DE-FC02-04ER15533. This is contribution number NDRL-5307 from the Notre Dame Radiation Laboratory. T.W.M. was additionally funded by Research Corporation for Science Advancement CCSA Award 7693, National Science Foundation-RUI Award 0809467, and the Benedictine University College of Science.

## Author contributions
T.W.M. and I.J. conceived the project, designed and performed the experiments, and carried out the data analysis. Both authors contributed equally in writing the manuscript.

## Competing interests
The authors declare no competing interests.
