## [Peer Review File · Communications Chemistry]

Reviewers' comments:

Reviewer #1 (Remarks to the Author):

The manuscript describes the VUV electronic absorption spectra of pressurized and supercritical CO₂. The reported striking phenomenon is that the diffuse vibrational structures in the VUV absorption spectrum gradually diminish with increasing pressure. The authors ascribe the structure changing to the perturbation of monomer CO₂ upper electronic PESs with the neighboring CO₂ molecules. The results are interesting, but some of statements are unclear. Some questions need to be addressed.

1. In Page 8, line 12, "...along the red edge of the VUV spectrum...", might be "the red edge"?

2. In Page 12, line 14-16, "Hence, one might conclude that the interaction energy should be at least 1.16 eV/0.27 eV = 4.3...". I am not sure the spectral shift for different molecules can compare directly, since the electronic states and spectrum sensitivity for CO₂ and H₂O should be different. It cannot simply deduce the interaction strength based on the electronic absorption spectrum shift.

3. In page 13, line 1-2, the fitting of the spectrum by using a simple Gaussian profile is a little arbitrary since only half spectrum has been recorded. The reference (JCP, 138, 224107(2013)) displayed the two parts of the spectrum around 148 nm and 133 nm. The Gaussian profile does not necessary to overlap with the spectrum backbone.

4. Is it possible the new species (CO₂ dimer with different structures) obscure the electronic absorption spectrum of CO₂ at high pressure?

Reviewer #2 (Remarks to the Author):

In this paper Marin and Janik report an experimental study of the absorption spectrum of supercritical CO₂ in the wavelength range 145-200 nm and at pressures between 19 and 137 bars. They report interesting results in a so far unexplored density regime, which have potential applications, among other fields, in atmospheric chemistry. The authors observe that the vibrational structures in the spectrum disappear gradually as the pressure is increased, and use a simple model to interpret these findings, suggesting that this observation can be understood as a perturbation of the potential energy surface of the excited states. This parallels a previous work done on supercritical water.

The paper is clear and well-written, but I am not fully convinced about the interpretation of the results. The manuscript would benefit from more details about the following points:

1. What is not clear is the effect of supercriticality. Could the authors comment on what the spectra would look like if CO₂ was not supercritical, i.e. by repeating the same study but at slightly lower temperature? This is discussed in the manuscript to some extent on p.12, but I find it very surprising that the authors observe absolutely no shift compared to the gas phase spectra. It seems that the supercriticality does not affect the spectrum at all.

2. The authors interpret their results in terms of critical intermolecular radii. For the highest pressures these radii become very small, and it is again surprising that this has only a very limited effect on the absorption spectrum. On p.19 the authors mention that the average critical radius is larger than the equilibrium distance of the CO₂ dimer. However, while this is true at the density

used for the discussion (0.224 g/cm³), this will not be the case at the highest pressure/density (0.767 g/cm³). Yet, this does not seem to have any effect on the spectrum. This fact raises some doubts about the interpretation of the authors. If the results can really be interpreted in terms of perturbation of excited states potentials, I would also expect a stronger impact of the pressure on the absorption spectra, rather than simply a progressive smoothing of the vibrational features as the pressure is increased.

Furthermore, at the highest pressures it is not clear to me that the behaviour of the system can be investigated only with a 2-body potential energy surface, as three-body effects should come into play.

I also have two minor comments:

- In Fig1 the meaning of the dashed lines should be defined in the caption.

- on L149: (0,0,0) is not defined

Reviewer #1 (Remarks to the Author):

The manuscript describes the VUV electronic absorption spectra of pressurized and supercritical CO₂. The reported striking phenomenon is that the diffuse vibrational structures in the VUV absorption spectrum gradually diminish with increasing pressure. The authors ascribe the structure changing to the perturbation of monomer CO₂ upper electronic PESs with the neighboring CO₂ molecules. The results are interesting, but some of statements are unclear. Some questions need to be addressed.

1. In Page 8, line 12, "...along the red edge of the VUV spectrum...", might be "the red edge"?

We apologize for this typographical error. It has been corrected.

2. In Page 12, line 14-16, "Hence, one might conclude that the interaction energy should be at least 1.16 eV/0.27 eV = 4.3...". I am not sure the spectral shift for different molecules can compare directly, since the electronic states and spectrum sensitivity for CO₂ and H₂O should be different. It cannot simply deduce the interaction strength based on the electronic absorption spectrum shift.

We thank the reviewer for this comment, as it is clear that we did not manage to clearly get our point across. What we are saying is the following. To a first approximation, except in instances with highly delocalized excited states, differences between the gas-phase vs. the solid-phase spectrum in most systems can be attributed to the absence vs. the presence of intermolecular forces and their impact on the observed spectral features and energy shift of the spectrum – how they affect the electronic states. We suggest that this is the case for CO₂, as we argued for H₂O in a previous publication. The CO₂ ice spectrum at a density of 1.60 g/cm³ is blue shifted by 0.27 eV compared to the gas-phase spectrum, and certainly intermolecular interactions must play a part in this energy shift. What we are arguing for our current data is that we observe ZERO spectral shift with increasing density in CO₂. Therefore the forces between CO₂ molecules must be much weaker than in H₂O and operating over much shorter distances. We have considerably revised and expanded upon the paragraph in question (see new pages 11-12) to clear this up for the reader.

3. In page 13, line 1-2, the fitting of the spectrum by using a simple Gaussian profile is a little arbitrary since only half spectrum has been recorded. The reference (JCP, 138, 224107(2013)) displayed the two parts of the spectrum around 148 nm and 133 nm. The Gaussian profile does not necessary to overlap with the spectrum backbone.

We absolutely agree with the reviewer that if one wanted to fit the entire spectrum, not just the spectrum witnessed in our accessible wavelength range, then a single Gaussian will not provide a good fit to the spectrum. However, a single Gaussian fits OUR data (a portion of the overall spectrum mentioned in the reference mentioned by the reviewer) perfectly well within the wavelength range that we measured. The point of the exercise in performing these fits was simply to provide a means to subtract out the spectral backbone within OUR experimental wavelength region and to isolate the vibrational structure. Based on Figure 4, clearly we managed to do so as the resulting residuals are flat.

4. Is it possible the new species (CO₂ dimer with different structures) obscure the electronic absorption spectrum of CO₂ at high pressure?

We are uncertain what the reviewer is implying by "obscure the electronic absorption spectrum." Yes, dimer formation will affect the spectrum. We dedicated several pages to discussing how the influence of dimer formation in various geometries might give rise to changes in the spectrum, invoking the dimer

potential energy surface for exploration of several possible geometries. However, we also pointed out that the known equilibrium constant for dimer formation it is highly unlikely that we are producing any substantial number equilibrated dimers at the pressures used in these experiments. Therefore, there are not enough dimers present to “obscure” the spectrum if this is what the reviewer is implying.

Reviewer #2 (Remarks to the Author):

In this paper Marin and Janik report an experimental study of the absorption spectrum of supercritical CO₂ in the wavelength range 145-200 nm and at pressures between 19 and 137 bars. They report interesting results in a so far unexplored density regime, which have potential applications, among other fields, in atmospheric chemistry. The authors observe that the vibrational structures in the spectrum disappear gradually as the pressure is increased, and use a simple model to interpret these findings, suggesting that this observation can be understood as a perturbation of the potential energy surface of the excited states. This parallels a previous work done on supercritical water.

The paper is clear and well-written, but I am not fully convinced about the interpretation of the results. The manuscript would benefit from more details about the following points:

1. What is not clear is the effect of supercriticality. Could the authors comment on what the spectra would look like if CO₂ was not supercritical, i.e. by repeating the same study but at slightly lower temperature? This is discussed in the manuscript to some extent on p.12, but I find it very surprising that the authors observe absolutely no shift compared to the gas phase spectra. It seems that the supercriticality does not affect the spectrum at all.

To our knowledge, the results that we present are the first of such pressure-dependent studies to be conducted on CO₂ in the VUV spectral region. We regret that we do not have data recorded at any subcritical temperatures, and we lack any apparatus to acquire further data at the moment. We are hopeful that with communication of our results in this manuscript will allow us to apply for the funding to further pursue these studies. Our highest accessible experimental pressure of CO₂ was limited by the supplier of CO₂ tank. In order to increase the pressure further we would need some sort of compressing apparatus. We note that we could increase density by studying pressurized liquid CO₂ at temperatures close to the triple point. However, the density of liquid CO₂ in the subcritical region does not differ much from the densities we already recorded in supercritical region (it is 0.72g/cc at the room-temperature coexistence point). Hence we did not expect to observe any appreciable difference. In order to perform more extended studies into the higher density region of liquid CO₂ (close to the triple point) we would need to entirely redesign our sample cell and vacuum chamber, which was not an option when the original experimental setup was established.

We agree with the reviewer that it is surprising to not find ANY spectral energy shift for our supercritical (most dense) CO₂ data compared to the gas phase. Therefore we made the argument that the ground state is not shifting to lower energies due to lack of dimer formation under our experimental conditions. However, we must insist that any appreciable expected shift would likely arise from tuning the pressure in our data to increase density to the values near the triple point of CO₂, and NOT be affected much by the higher temperature needed to achieve supercritical conditions. We refer the reviewer to temperature-dependent measurements by Vernot, *et al.* that we referenced in this regard. In our similar studies with VUV spectroscopy of supercritical H₂O, the nature of spectral features below or above the critical temperature is not akin to behavior on either side of a true phase transition. Nor would we

expect this for CO₂, or any other fluid. For a supercritical fluid, the density is continuously tunable from gas-like up to liquid-like densities, giving a wide range of conditions to observe the onset and influence of neighboring molecules as the distance between them changes. At lower temperatures such tunability exists only for the gas phase, which can be manipulated up to the critical pressure – 73.8 bar for CO₂. In our data, we witness the gradual loss of vibrational structure at pressures well below 73.8 bar. Thus, we imagine that for lower-temperature CO₂ we would see similar gradual losses in vibrational structure for the gas phase with increasing pressure, without the presence of a liquid phase. We expect that formation of the liquid would be reflected by a sudden change in the spectrum. We have now included this in the text on page 12 and in the conclusions regarding potential future work to explore once we construct a new in-house VUV spectrometer and develop capabilities to explore VUV spectrum of liquid CO₂ near its triple point and densities comparable to the solid phase.

2. The authors interpret their results in terms of critical intermolecular radii. For the highest pressures these radii become very small, and it is again surprising that this has only a very limited effect on the absorption spectrum. On p.19 the authors mention that the average critical radius is larger than the equilibrium distance of the CO₂ dimer. However, while this is true at the density used for the discussion (0.224 g/cm³), this will not be the case at the highest pressure/density (0.767 g/cm³). Yet, this does not seem to have any effect on the spectrum. This fact raises some doubts about the interpretation of the authors. If the results can really be interpreted in terms of perturbation of excited states potentials, I would also expect a stronger impact of the pressure on the absorption spectra, rather than simply a progressive smoothing of the vibrational features as the pressure is increased.

The arguments we can use here are essentially the same we raised in the previous point. The fact that there is no change in the spectral position, which we assume is the issue reviewer is raising here, relates to the fact that even at the density of 0.767 g/cm³ at 38 °C and close proximity of neighboring CO₂ molecules, we are not forming enough dimers to experience their effect on the spectral position. The interaction is clearly not as strong as the reviewer's intuition suggests. The potential energy of attraction scales as approximately $1/r^6$, so the attraction does not become substantial until close proximity of molecules is achieved. If experiments at much higher pressures were to be performed one might see these effects come to play. Again, we have proposed this for future work.

Furthermore, at the highest pressures it is not clear to me that the behaviour of the system can be investigated only with a 2-body potential energy surface, as three-body effects should come into play.

The reviewer is correct in this regard. Certainly, at our highest pressures one might need to include 3-body interactions. To our knowledge, a CO₂ trimer potential energy surface does not exist in the published literature. As such, we have done our best with the high-quality dimer potential that is available.

I also have two minor comments:

- In Fig1 the meaning of the dashed lines should be defined in the caption.

We were remiss in making this definition and have fixed the caption.

- on L149: (0,0,0) is not defined

The autocorrelation function is the scalar product of the initial wavepacket at time $t = 0$ and the evolving wavepacket at time t , establishing the interrelation between the time and the frequency domain. It is the link between the time-dependent molecular dynamics and the absorption spectrum, as the spectrum is the Fourier transform of the autocorrelation function, where $(0,0,0)$ represents the system coordinates prior to excitation. We have revised the text accordingly on page 6 to better explain this.

REVIEWERS' COMMENTS:

Reviewer #1 (Remarks to the Author):

The current version addresses all my questions. In my opinions I recommend it to be considered for publication.

Reviewer #2 (Remarks to the Author):

I thank the authors for their additional comments. In my opinion the paper can be accepted for publication.